# Improved Particle Swarm Optimization for Sea Surface Temperature Prediction

**Qi He [1], Cheng Zha [1], Wei Song [1,\*]⬤, Zengzhou Hao [2], Yanling Du [1], Antonio Liotta [3]⬤ and Cristian Perra [4]⬤**

1   Department of Information Technology, Shanghai Ocean University, Shanghai 201306, China; qihe@shou.edu.cn (Q.H.); m170550834@st.shou.edu.cn (C.Z.); yldu@shou.edu.cn (Y.D.)
2   State Key Laboratory of Satellite Ocean Environment Dynamics, Second Institute of Oceanography, Ministry of Natural Resources, Hangzhou 310012, China; hzyx80@sio.org.cn
3   School of Computing, Edinburgh Napier University, Edinburgh EH10 5DT, UK; a.Liotta@napier.ac.uk
4   Department of Electrical and Electronic Engineering, University of Cagliari, Via Marengo, 2, 09100 Cagliari, Italy; cperra@ieee.org
\*   Correspondence: wsong@shou.edu.cn

**Abstract:** The Sea Surface Temperature (SST) is one of the key factors affecting ocean climate change. Hence, Sea Surface Temperature Prediction (SSTP) is of great significance to the study of navigation and meteorology. However, SST data is well-known to suffer from high levels of redundant information, which makes it very difficult to realize accurate predictions, for instance when using time-series regression. This paper constructs a simple yet effective SSTP model, dubbed DSL (given its origination from methods known as DTW, SVM and LSPSO). DSL is based on time-series similarity measure, multiple pattern learning and parameter optimization. It consists of three parts: (1) using Dynamic Time Warping (DTW) to mine the similarities in historical SST series; (2) training a Support Vector Machine (SVM) using the top-k similar patterns, deriving a robust SSTP model that offers a 5-day prediction window based on multiple SST input sequences; and (3) developing an improved Particle Swarm Optimization (PSO) method, dubbed LSPSO, which uses a local search strategy to achieve the combined requirement of prediction accuracy and efficiency. Our method strives for optimal model parameters (pattern length and interval step) and is suited for long-term series, leading to significant improvements in SST trend predictions. Our experimental validation shows a 16.7% reduction in prediction error, at a 76% gain in operating efficiency. We also achieve a significant improvement in prediction accuracy of non-stationary SST time series, compared to DTW, SVM, DS (i.e., DTW + SVM), and a recent deep learning method dubbed Long-Short Term Memory (LSTM).

**Keywords:** sea surface temperature; sea surface temperature prediction; similarity measure; support vector machine; particle swarm optimization; local search

## 1. Introduction

Sea Surface Temperature (SST) is one of the crucial factors affecting the ocean climate. The occurrence of events such as El Niño, storm surges and red tides are closely related to SST. Therefore, in recent years, Sea Surface Temperature Prediction (SSTP) has attracted more and more attention in various marine-related fields such as marine meteorology, navigation, marine disaster prevention and mitigation, and marine fisheries. So far, researchers across the world have proposed many methods to predict SST, which can be divided into three categories: statistical forecasting [1], numerical forecasting [2], and empirical forecasting [3].

SST data are long-term data sequences and typically involve large data volumes. Many scholars regard SSTP as a time-series regression problem, and derive prediction models by fitting the curve

of historical data [4]. However, SST data is well-known to suffer from high levels of redundant information, which makes it very difficult to realize accurate predictions through time-series regression, nor to capture the complex dynamics of SST trends. In 2013, Lins et al., provided a prominent study showing how classic Support Vector Machine (SVM) could be used to predict SST in terms of raw data, slope and curvature [5]. Yet, they showed that SVM exhibited very similar performance when using raw data and curvature features, and was more effective than using slope. Better prediction accuracy was achieved with the more recent Long Short-Term Memory (LSTM) method [6]. This type of neural network has been widely used in diverse areas, thanks to its suitability for processing time series. In fact, Zhang et al. [7] were the first to use LSTM to predict SST. First, the SST sequence features were learned by LSTM layers. Then, a fully-connected layer was used to map the output of the LSTM layers to the final prediction result. Both SVM and LSTM tackle prediction accuracy. Yet, neither method is sufficiently efficient. Even more critically, marine operational forecasting requires significant improvements in SST trend predictions. Typically, empirical methods are used, of which the Analog Complexing (AC) algorithm [8] is a representative example. Nevertheless, AC has not shown sufficient accuracy in practical applications [9].

In this paper, our aim is to propose a new SSTP method, which not only can predict SST rapidly and accurately, but also can better model the SST trend. The main idea of our method is to use similarity measure to mine historical SSTs in order to extract sequences having similar trends, and then feed these into a suitable prediction method to infer future SSTs.

When using this method to perform SSTP, we need to consider two important issues. First, time-series mining is critically sensitive to the accuracy of the similarity measure method; thus, any unexpected error in the mined data will dominate the prediction result. We solve this problem by a combination of similarity measure and multiple time-series regression.

Another criticality is introduced by the choice the regression model parameters, which have a dramatic impact on the performance of the prediction model. We tackled this issue as a bi-objective optimization problem. In fact, our solution is dubbed DSL, since it is evolved from three building blocks: Dynamic Time Warping (DTW) [10], SVM [11], and Local-search enabled parameters optimization [12]. It consists of three parts:

1. Using Dynamic Time Warping (DTW) to mine the similarities in historical SST series. DTW has been chosen as the result of an experimental, comparative analysis of three representative time-series SST similarity methods. It led to the highest prediction accuracy, it better modeled the SST trends, and was found to be suitable to mining SST long-term time series.

2. Training a Support Vector Machine (SVM) using the top-k similar patterns, deriving a robust SSTP model that offers a 5-day prediction window based on multiple SST input sequences. Learning from multiple time-series sequences was instrumental to facilitating consistency enhancement and noise cancellation, thus achieving high prediction accuracy.

3. Developing an improved Particle Swarm Optimization (PSO) method, dubbed LSPSO, which uses a local search strategy to achieve the combined requirement of prediction accuracy and efficiency. We were striving for optimal model parameters, to pursue SST prediction efficiency, providing a new way for marine operational forecasting.

Overall, our work provides a new SSTP method that not only improves prediction accuracy and speed, but also better models the trend of SST changes. An important element was to find a suitable method to optimize the parameters of the combined DTW + SVM prediction method (dubbed DS hereafter). We achieved this goal by developing an improved Particle Swarm Optimization (PSO) method, which was responsible for a 16.7% improvement in prediction accuracy, in terms of reducing the Root Mean Square Error (RMSE), and for a 76% reduction in prediction time. We also achieved a significant improvement in prediction accuracy of non-stationary SST time series, compared to DTW, SVM, DS (i.e., DTW + SVM), and a recent deep learning method dubbed LSTM (Long-Short Term Memory).



The rest of this paper is organized as follows. In Section 2, we describe related work on SSTP, similarity measures, and optimization methods. The main idea of DSL is introduced in Section 3. Experimental results and performance evaluation are shown in Section 4. Section 5 presents our conclusions.

## 2. Related Work

SST data is typically long-time-sequence data, hence many researchers have regarded SSTP as a time-series regression problem, thus applying time-series prediction methods to SSTP. The traditional time-series prediction methods such as Autoregressive (AR) [13], Moving Average (MA) [14] and Autoregressive Moving Average (ARMA) [15] are linear. Yet, SST has non-stationary and nonlinear characteristics, thus these linear methods are not well-suited to the practical application of SSTP.

Therefore, researchers have proposed some nonlinear methods to predict SST. Li et al. [16] have used SVM to predict SST and achieved good results. AC is often used for hydrological forecasting, and can also be used for time-series prediction. It can better mine the hidden information in the sequence, yet the method is sensitive to factors such as Pattern Length (PL), Interval Step (IS), and similarity measure.

In the process of time-series data mining, the similarity measure is the basis of clustering, association rules and prediction. Euclidean distance [17] is a commonly used time-series similarity measure. It is simple to calculate, but can only process time series of equal length. It cannot handle the sequence stretching and bending on the time axis.

DTW is based on the idea of Dynamic Programming (DP) [18], which was originally applied in isolate word speech recognition [19]. It was then introduced into the study of similarity measures of time series by Berndt et al. [20], achieving good results. DTW overcomes the shortcomings of Euclidean distance. It will not only measure the similarity of time series of different lengths, but also supports the stretching and bending of sequences on the time axis. It has a good measurement accuracy and robustness, which is why it is widely used.

Cosine similarity [21] uses the cosine of the angles of two vectors in the vector space to measure the similarity between them. The closer the cosine value is to 1, the closer the angle is to 0, the more similar they are. Therefore, the method can better distinguish the difference in the direction of differentiation, and is not sensitive to the cosine value.

By comparing the evaluation index of the different similarity measures, one that is suitable for measuring the similarity of SST sequences can be selected, and the multi-objective optimization method can be used as a solution to select the appropriate values for PL and IS.

Many scholars have studied the theory and application of multi-optimization algorithms. Differential Evolution (DE) [22] is a simple but efficient parallel search algorithm proposed by R. Storn in 1997, whose principle is similar to genetic algorithms. Zitzler et al. [23] compared the Non-dominated Sorting Genetic Algorithm (NSGA) [24] with the Niche Pareto Genetic Algorithm (NPGA) [25], and the Vector-Evaluated Genetic Algorithm (VEGA) [26], finding that NSGA has the best performance. Yet, the computational complexity of NSGA is higher.

Aiming at the shortcomings of NSGA, Deb et al. [27] proposed the Elitist Non-dominated Sorting Genetic Algorithm (NSGA-II), which reduces the time complexity of Pareto dominating sorting [28]. It is suitable for dealing with low-dimensional, multi-objective optimization problems. However, when dealing with high-dimensional, multi-objective problems, the crowding distance [29] is not applicable in high-dimensional space, and the computational complexity is also high.

Inspired by the predatory behavior of natural bird populations, Kennedy [30] proposed Particle Swarm Optimization (PSO). PSO is an evolutionary computation method based on individual improvement, population cooperation and competition mechanisms. It has the important characteristics of being simple and easy to implement, which makes it suited to single-objective optimization. When optimizing multiple targets (as in MOPSO [31]), M. Reyes et al. proposed OMOPSO (Optimal Multi-Objective Particle Swarm Optimization) [32], which uses the crowding factor to select the leaders,

based on Pareto dominance. Other researchers proposed hybrid optimization strategies that combine two or more heuristic optimization techniques [33,34], which could integrate the merits of different algorithms or achieve a near optimal solution quickly.

Considering that DS will fall into local optimum when optimizing the SSTP model, we propose LSPSO (Particle Swarm Optimization algorithm combined with Local Search strategy), which uses the Pareto dominance relationship to measure the advantages and disadvantages of the solution, and uses local search for non-dominated solution sets to enhance the local search ability. LSPSO has a strong ability to explore and can approach the true Pareto frontier.

## 3. The Proposed DSL Method

Our method involves three main algorithms, DTW, SVM, and LSPSO, and we have dubbed it DSL for short. Given an SST sequence $F = F_1, F_2, \ldots, F_{|F|}$ (whereby $|F|$ is the length of the sequence $F$), our aim is to predict the SST for the subsequent five days. Figure 1 shows the flow chart of our SSTP algorithm based on the similarity measure.

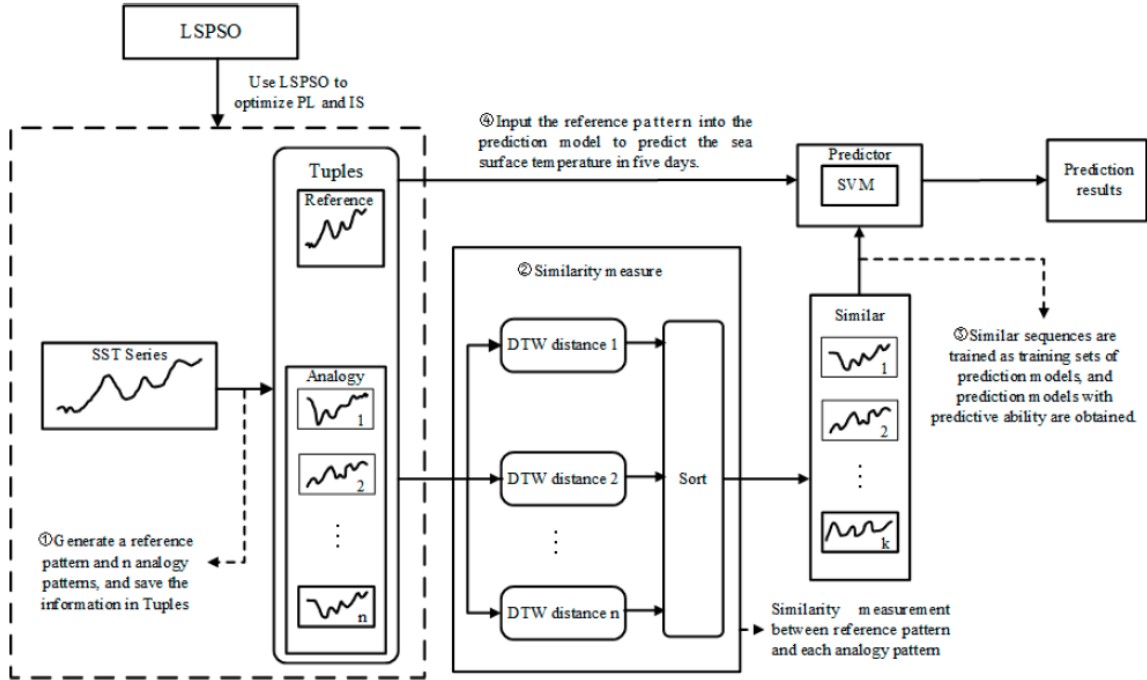

**Figure 1.** Flow chart of SSTP algorithm based on similarity measure.

The main steps of DSL are as follows: (1) Read the SST sequence $F$, generating a reference pattern and n analog patterns, and storing the information in the tuples T. (2) Calculate the DTW distances of the reference pattern and each analog pattern separately, and sort them in ascending order according to the DTW distance; the first k modes can be regarded as analog patterns similar to the reference pattern. (3) The obtained analog patterns are used as the input of the SVM model, and the corresponding SST of the following five days in the sequence $F$ are used as the output of the SVM model; an SVM model with predictive ability is obtained through training. (4) The reference pattern is used as the input to the SVM model, generating a 5-day prediction of the SST. 

The reference pattern represents the current SST trend, while the analog pattern is the historical SST trend. In the process of generating the reference and the analog patterns, the PL and IS are uncertain; so before using DSL to predict SST, it is necessary to use LSPSO to optimize the model parameters and obtain the appropriate PL and IS.

### 3.1. Generation and Trend Prediction

In order to mine the similar trend to the current SST from the historical SST, we need to generate the reference and the analog patterns. The reference pattern represents the recent trend, whereas the analog pattern represents the historical trend. Based on the persistence and similarity of SST, we can calculate the similarity between reference pattern and analog pattern to mine the historical trends that are close to the current trends, and then use historical trends to predict current trends.

Algorithm 1 specifies the process of generating the reference and the analog patterns. Step 2 takes out the last five days of $F$ for evaluating the prediction model; step 3 generates the reference pattern C; steps 4–9 generate the analog pattern A and save the reference and the analog patterns in the tuples T.

---

**Algorithm 1.** Generating the reference pattern C and the analog pattern A.

---

**Input:**
SST sequence: $F$; PL: m; IS: step;
**Output:**
       Tuples: T;
1:   T←Φ; A←Φ; C←Φ; D←Φ; Q←Φ;
2:   Take the last five days of $F$ as the true value, and remove the last five days of $F$ to obtain the sequence D;
3:   Take the SST of the last m days of D as the reference pattern C, and remove the SST of the m days to obtain the sequence Q;
4:   t = 1; // t record the starting position of each analog pattern
5:   while ((t + m − 1)<(|F| − m − 5)) do
6:        Take the SST of the sequence Q from t to t + m - 1 as the analog pattern A;
7:        Save the generated reference and analog modes in the tuples T;
8:        t = t + step;
9:   End While

---

The similarity of the SST sequence is measured by the DTW distance, which can be performed in two steps: firstly, calculate the distance matrix by calculating the distance between the SST points of A and C; then, find an optimal path in the matching distance matrix. $C = c_1, c_2, \ldots, c_m$ and $A = a_1, a_2, \ldots, a_m$ define a matrix of m rows and m columns, with elements $d(a_i, c_i) = |a_i - c_i|$. W is the distance matrix between A and C, and can be expressed as per Equation (1).

$$
W = \begin{bmatrix}
d(a_1, c_1) & \cdots & d(a_1, c_m) \\
d(a_2, c_1) & \cdots & d(a_2, c_m) \\
\vdots & \ddots & \vdots \\
d(a_m, c_1) & \cdots & d(a_m, c_m)
\end{bmatrix}
\tag{1}
$$

In the distance matrix W, the similarity relationship between A and C is represented by a set of consecutive matrix elements, which is referred to as a curved path L, $L = (w_1, w_2, \ldots w_K)$, where $m \leq K \leq 2m - 1$.

Path selection needs to meet the following constraints: (1) The start and end points of A and C are aligned, whereby the starting point of the path is $(a_1, c_1)$, and the ending point is $(a_m, c_m)$. (2) Any point on the path can only move along the adjacent elements of the matrix each time. That is, if $w_1 = (a_1, c_1)$ then next point $w_2 = (a, c)$ for the path, to satisfy $a - a_1 \leq 1$ and $c - c_1 \leq 1$. (3) Any point on the path can only move one way along the time axis each time. That is, if $w_1 = (a_1, c_1)$ then next

point $w_2 = (a, c)$ for the path to satisfy $a - a_1 \geq 0$ and $c - c_1 \geq 1$. The shortest path distance that satisfies the above constraints is the DTW distance, which can be expressed by Equation (2).

$$DTW(A, C) = \min\left\{ \sum_{k=1}^{K} w_k \right\} \tag{2}$$

Through the above steps, the DTW distance between each analog pattern and the reference pattern can be calculated. The smaller the DTW distance, the more similar the two patterns are. So far, we have obtained analog patterns similar to reference patterns from $F$, showing how to use these analog patterns to predict the SST for the next five days. AC is calculating the average of the SST for the next five days that corresponds to these analog patterns in $F$ as the SST prediction value for the next five days.

Yet, this method does not make full use of the information obtained through data mining, which leads to low prediction accuracy. In response to this problem, this paper uses these analog patterns to establish prediction modes. Because these analog patterns have a small sample size, they can only be modeled using small sample prediction methods to maximize information utilization. Although Back Propagation (BP) neural networks [35] can be used for small-sample prediction, model training is difficult and prone to failing during the training process. SVM is suitable for processing small samples and nonlinear problems, and has good robustness and high prediction accuracy. Considering that SST has nonlinear features, this paper combines SVM with DTW to construct a predictive model. In this paper, the analog patterns similar to the reference patterns are the input of the SVM, and the SST of the next five days corresponding to these analog patterns in F is used as the output of the SVM. Through training, we obtain a predictive SVM model and then use the reference model as the model input to get the SST for the next five days.

The training samples of the SVM are $(X_i, Y_i)$, $i = 1, 2, ..., k$, where $X_i \in R^m$ is the input, $Y_i \in R^n$ is the output, and $m$ and $n$ are the dimensions of the variable, and $k$ is the number of samples. The goal of the SVM is to build the following regression function:

$$f(X) = W\varphi(X) + b = \begin{bmatrix} W_1\varphi(X) + b_1 \\ \vdots \\ W_n\varphi(X) + b_n \end{bmatrix} \tag{3}$$

$\varphi(X)$ can map data from the original space to the high-dimensional space, which can transform the nonlinear problem in the original space into a problem that can be solved in high-dimensional space. $W$ and b are weights and offsets, respectively. To determine $W$ and $b$ we proceed with the minimization (4).

$$\min \frac{1}{2} \sum_{j=1}^{n} \|W\|^2 + C \sum_{j=1}^{n} \sum_{i=1}^{k} L_j(f_j(X_i), Y_i^j) \tag{4}$$

where $C$ is the balance factor, $L_j()$ is the loss function, and $j$ is the dimension of the output variable. The loss function is usually defined as follows:

$$L_j(f_j(X_i), Y_i^j) = \begin{cases} 0, f_j(X_i) - Y_i^j < \varepsilon_j, \varepsilon_j > 0, \\ f_j(X_i) - Y_i^j - \varepsilon_j, otherwise. \end{cases} \tag{5}$$

The multivariate nonlinear regression SVM model can be finally expressed by the Lagrangian method as:

$$f_j(X) = \sum_{i=1}^{k} (a_i^j - a_i^{j*})K(X_i, X) + b_j \tag{6}$$

where $a_i^j, a_i^{j*}$ are Lagrangian multipliers, and $K(X_i, X)$ is a kernel function. RBF is the most widely used kernel function, which not only can implement nonlinear mapping, but also has fewer parameters. Therefore, this paper uses RBF as the kernel function of SVM. By training the above model, an SVM with predictive ability can be obtained, and then, inputting the reference pattern into the model, we can predict the SST for the next five days.

Although SSTP can be implemented in this way, the PL and IS may dramatically affect the accuracy and efficiency of the DS. In order to determine the appropriate PL and IS, this paper proposes LSPSO and applies it to model parameter optimization.

### 3.2. Parameter Optimization Using LSPSO

The basic operations of a standard PSO are particle speed and position updates:

$$v_{i,d}^{k+1} = wv_{i,d}^k + c_1 r_1 (p_{Pbest,d}^k - x_{i,d}^k) + c_2 r_2 (p_{Gbest,d}^k - x_{i,d}^k) \tag{7}$$

$$x_{i,d}^{k+1} = x_{i,d}^k + v_{i,d}^k \tag{8}$$

where w is the inertia weight, $c_1, c_2$ are learning factors, $r_1, r_2$ are mutually independent random numbers in the interval [0, 1], $k$ is the number of evolutions, $d$ is the search space dimension. Pbest is the optimal location for each particle search, and Gbest is the optimal location for the entire particle swarm search.

When PSO optimizes the parameters of the SST model, the search ability is weak in the late search, and it is easy to fall into local optimum, which is not conducive to optimize the model parameter. Therefore, we proposed LSPSO, which differs from the PSO as summarized below:

(1) PSO adopts a random method in population initialization, which leads to each particle appearing in a random distribution state in space, lacking the guidance of prior knowledge, which is not conducive to the particle close to the optimal solution. In this paper, the Beta distribution strategy is used to initialize the population, which is beneficial to the rapid formation of the surrounding situation by the particles to the optimal solution.

(2) The global and local search capabilities of PSO are mutually constrained and tend to fall into local optimum in the later stages of search. This paper uses local search strategy to enhance the local search ability of PSO, so that the improved PSO has independent global and local search capabilities.

(3) Particles tend to cross out of bounds during flight. Particles flying faster than the bound range will mutate; flying speeds that exceed half of the constraint range and below the constraint range decelerate to prevent particles from crossing the boundary.

When LSPSO optimizes the parameters of DS, it can be expressed as follows:

$$
\begin{aligned}
&\min \text{RMSE} = G_1(PL, IS) \\
&\min \text{RT} = G_2(PL, IS) \\
&\qquad\qquad s.t \\
&PS = \{x | 1 \le x \le 360, x \in Z\} \\
&IS = \{y | 1 \le y \le 30, y \in Z\}
\end{aligned} \tag{9}
$$

where RMSE gives the accuracy of the DS; RT is the running time of the DS; PL represents the pattern length; and IS represents the interval step. Figure 2 depicts the flow chart of LSPSO, including the following steps:

**Step 1:** Randomly initialize the population using the Beta strategy; each particle has two attributes: the pattern length and the interval step;

**Step 2:** Calculate the RMSE and RT of each particle, and then filter out the non-dominated solution according to the non-dominated relationship, and store it in the external population, S;

**Step 3:** Local search of the external population, S by local search, to enhance the local search ability of the particle, and obtain the population S';

**Step 4:** Control the population in population D by crowding distance;

**Step 5:** Update Pbest and Gbest;

**Step 6:** Update the velocity and position of the particle according to Formulas (7) and (8);

**Step 7:** Determine whether the termination condition is met. If it does, output the optimal solution set and select the appropriate parameters for predicting SST; otherwise, add G to 1, and return to step 2.

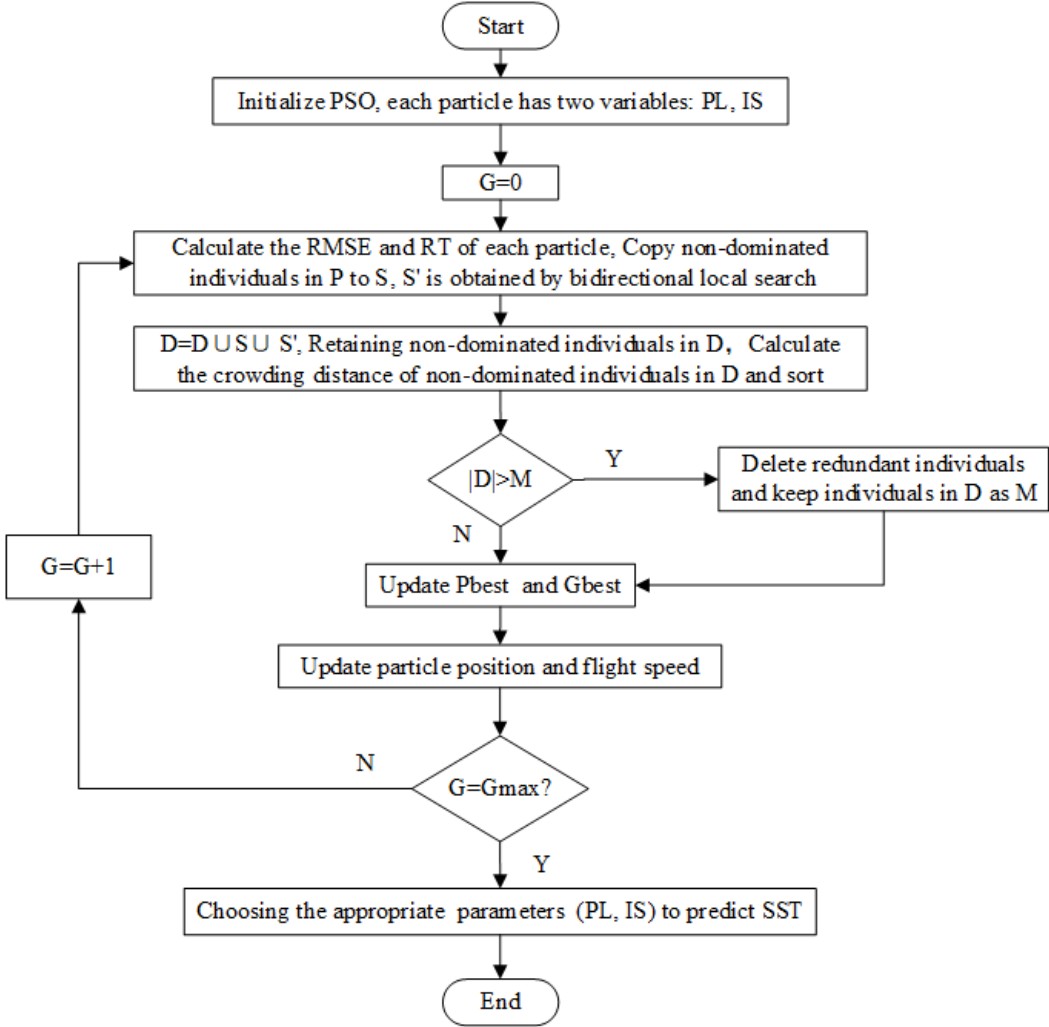

**Figure 2.** Flow chart of LSPSO.

The traditional particle swarm optimization algorithm typically uses random initialization to generate the initial solution. Due to the lack of guidance of prior information, it is not conducive to the initial particle to move closer to the optimal solution. In this paper, the Beta distribution initialization strategy is used to initialize the population, which is beneficial to the particle to form the enclosing situation to the optimal solution. The shape of the Beta function is a symmetric U-shape, and the candidate set is most likely to be located near the boundary of the search space. At this time, the global optimal solution is better enclosed within the initial particle population. The Beta distribution function is defined as:

$$\beta(x; m, n) = \frac{x^{m-1}(1-x)^{n-1}}{B(m,n)}, 0 < x < 1. \tag{10}$$

The denominator is defined as follows:

$$B(m,n) = \int_0^1 t^{m-1}(1-t)^{n-1}dt \tag{11}$$

Pareto dominance relationships are used in many optimization algorithms to measure the pros and cons of a solution, usually defined as follows:

**Definition 1.** *(non-dominated relationship). Let x, y be the two decision variables in equation (12), if x dominates y, then if and only if:*

$$\forall i \in \{1,2,\ldots,n\}, \exists j \in \{1,2,\ldots n\}; f_i(x) \leq f_i(y) \cap f_j(x) < f_j(y). \tag{12}$$

**Definition 2.** *(non-dominated solution). Assuming that there is a solution set P, where individual q is not dominated by any other individual, then q is a non-dominated individual in P. The subset of non-dominated individuals of P is called the non-dominated set (NDset) of P.*

$$NDset = \{q|q \in P \text{ and does not exist } p \in P, \text{make } p \text{ dominate } q\}$$

Since only the non-dominated solution in the population P needs to be selected, in the process of comparing different particles, once the particles are dominated by other particles, they are not compared with other particles, which reduces the complexity of the algorithm. Through the non-dominated relationship, a non-dominated solution set S can be obtained. The solutions contained in the set are non-dominated solutions, and some better solutions can be found around the non- dominated solution.

The local search ability of the algorithm is enhanced by the local search strategy, which is beneficial to improve the local search ability of the algorithm. For the non-dominated solution set S, one of the individuals is $x_{i,k} = (x_{1,i,k}, x_{2,i,k}, \ldots x_{n,i,k})^T$, $i$ represents the $i$-th individual in S, $k$ is the number of evolutions, and $n$ represents the dimension of the variable. The search for the $m$-th variable of the individual $x_{i,k}$ in two directions can be expressed below as $L_{m,i,t}$ and $R_{m,i,t}$.

$$L_{m,i,t} = x_{m,i,t} - c \times |P_{m,i,t} - G_{m,i,t}| \tag{13}$$

$$R_{m,i,t} = x_{m,i,t} + c \times |P_{m,i,t} - G_{m,i,t}| \tag{14}$$

where P and G are randomly selected from the non-dominated solution set S, and c is the interference coefficient. New individuals can be generated by Equations (13) and (14). Algorithm 2 expresses the complete process of local search.

The non-dominated solution set S obtained from each evolution, the local searched population S′ and the external archive D, are merged into a new external archive D. The non-dominated solutions in D are used as external archives to ensure the diversity of the population.

As the number of iterations increases, the size of the external archive will gradually increase too. It is unrealistic to include all non-dominated solutions in D. At this time, it is feasible to obtain a finite non-dominated subset that can represent the solution space. In order to obtain a more uniform non-dominated solution set, this paper uses the crowded distance to remove redundant individuals and controls the number of non-dominated solutions in D.

After updating Gbest and Pbest, we use Formulas (7) and (8) to update the velocity and position of the particle. Since the particles will cross the boundary when flying, the particles will be mutated for the particles whose particle velocity exceeds the constraint range. The formula is as follows:

$$M_{n,i,t} = E_{n,i,t} \times (1 - rand\,(-0.2, 0.2)) \tag{15}$$

where E is randomly selected from the external archive D, and M is the mutated particle. When the flying speed of the particle exceeds half of the constraint range, but is still below the constraint range, the particle velocity is set to half of the constraint range to prevent the particles from crossing the boundary. Through iteration, if the termination condition is met, the optimal solution set can be output finally, and a reasonable parameter combination is selected from the solution set as a parameter of the prediction model for predicting the SST for the next five days.

---

**Algorithm 2.** Local Search.

---

**Input:**
　　　The non-dominated solution set: S;
　　　Number of non-dominated solutions: |S|;
　　　The dimension of the search space: *n;*
**Output:**
　　　External population: S';
1:　S'←Φ;
2:　For I = 1 to |S| do
3:　　　Randomly select two individuals P and G from S
4:　　　For j = 1 to n do
5:　　　　　Generate individual L and R by Equations (13) and (14)
6:　　　End For
7:　　　Keep better individuals in both L and R in S'
8:　End For

---

## 4. Results and Discussion

### 4.1. Experimental Environment and Data

The experimental environment of this paper has the following specifications: Windows 10 operating system; Intel Core i5 CPU; 2.6 GHz clock; and 8Gbyte RAM. The algorithms MODE, NSGA-II, and OMOPSO are implemented using Matlab, while the other components have been done in Python. The experimental data includes 9 SST sequences provided by the Second Institute of Oceanography, including SSTs from 1 January 2004 to 31 December 2016.

### 4.2. Evaluation Indicators and Test Functions

In this paper, we used three indicators to evaluate our model performance. We used Root Mean Square Error (RMSE) [36] to assess the overall accuracy of the SSTP models. To evaluate the convergence and uniformity of the parameter optimization algorithm, we used Generational Distance (GD) [37] and Spacing (SP) [38] as the indicators, and introduced three test functions. The details of these indicators and test functions are described in the following.

RMSE is a measure of the deviation between the observed value and the true value. The smaller the value, the more accurate the prediction. The formula of RMSE is as follows:

$$RMSE = \sqrt{\frac{\sum_{i=1}^{n}\left(Y\_real_i - Y\_pred_i\right)^2}{n}} \tag{16}$$

where *Y_real* is the true value, *Y_pred* is the predicted value, and n is the number of days to predict.

The Pareto optimal solution set obtained by the multi-objective optimization algorithm should maintain the convergence of the solution and the uniformity of the distribution. In order to evaluate the convergence and uniformity of the Pareto frontier obtained by the multi-objective optimization algorithm, to obtain as many Pareto optimal solutions as possible, and to approach the true Pareto

frontier as best as possible, GD was used as the solution convergence performance evaluation. The smaller the value of GD, the better the convergence of the solution set. Secondly, the Pareto optimal solution should be evenly distributed along the Pareto frontier as much as possible, and SP was used as the index of uniform distribution performance evaluation. The smaller the SP, the more uniform the solution set distribution. GD is defined as:

$$GD = \sqrt{\sum_{i=1}^{n} \frac{d_i^2}{n}} \tag{17}$$

where n is the optimal number of Pareto solutions, and $d_i$ is the distance of the i-th Pareto optimal solution in the objective space from the nearest individual of the Pareto frontier. SP is defined as:

$$SP = \sqrt{\frac{1}{n-1}\sum_{i=1}^{n} (\bar{d} - d_i)^2} \tag{18}$$

where n is the optimal solution number of Pareto, $d_i$ is the distance of the i-th Pareto optimal solution from other individuals in the objective space, and $\bar{d}$ is the average value of $d_i$.

LSPSO was applied to the bi-objective optimization of accuracy and efficiency of the SSTP. In order to verify the feasibility of LSPSO in the bi-objective optimization problem, three commonly used bi-objective test functions were selected for testing: BNH [39], SRN [40] and TNK [41]. We compared the GD and SP indicators of the optimal frontier obtained by MODE, NSGA-II, and OMOPSO, respectively. Table 1 lists the characteristics of these three test functions.

**Table 1.** Bi-objective optimization problems.

| Function | Number of Variables | Number of Objectives | Analytical Pareto Frontier |
|---|---|---|---|
| BNH | 2 | 2 | Connected/Convex |
| SRN | 2 | 2 | Disconnected/Convex |
| TNK | 2 | 2 | Disconnected/Nonconvex |

*4.3. Analysis of Experimental Results*

In this work we designed three sets of experiments: (1) to compare the advantages and disadvantages of the similarity measures in SSTP and choose the best method; (2) to verify the effectiveness of LSPSO by comparing it with the MODE, NSGA-II, and OMOPSO algorithms; (3) to compare the performance of DSL with other SSTP methods.

**Experiment 1: Comparison of similarity measures.** *Applying AC to SSTP requires choosing a suitable similarity measure method to measure the similarity of SST sequences. Therefore, we first computed the Euclidean distance, the cosine distance, and the DTW distance to measure the similarity of SSTs. Then, we chose the optimal similarity measure based on this principle: the better the similarity is measured, the smaller the SSTP error is. The error of SSTP was measured by Root Mean Square Error (RMSE).*

Table 2 shows the achieved RMSE in relation to SST predictions based, respectively, on Euclidean distance, Cosine distance, and DTW distance similarity measures. The first nine rows provide RMSE values for nine different SST sequences, with the average values given in the last row. When using the Euclidean distance to predict SST, the average RMSE is 0.4949, which is slightly higher than (at times comparable to) DTW, but much better than Cosine. The reason for this is that the number of days in which the SST changes regularly is not fixed, and the Euclidean distance does not support scaling. However, the DTW distance can better overcome this deficiency, so DTW can better reflect the similarity of SST changes.

**Table 2.** RMSE obtained by predicting SST with three similarity measures.

| Station | Euclidean | Cosine | DTW |
|---|---|---|---|
| 1 | 0.5289 | 1.3223 | 0.4866 |
| 2 | 0.5525 | 1.2907 | 0.5136 |
| 3 | 0.4813 | 0.8982 | 0.4606 |
| 4 | 0.4641 | 0.6669 | 0.4637 |
| 5 | 0.4788 | 1.4734 | 0.4544 |
| 6 | 0.5247 | 0.6666 | 0.4630 |
| 7 | 0.5364 | 0.5549 | 0.5204 |
| 8 | 0.4437 | 1.3070 | 0.4429 |
| 9 | 0.4439 | 1.4096 | 0.4161 |
| Avg | 0.4949 | 1.0655 | 0.4690 |

The average RMSE when using the cosine distance to predict SST is 1.0655. This is much higher than the other two indexes, leading to a much worse prediction ability. This happens because the cosine distance uses the cosine of the angle of the SST vector to measure its similarity, which only reflects the trend of SST changes, and is not sensitive to the value of the SST itself. The DTW and Euclidean distances are based on the SST values. In summary, DTW has the highest prediction accuracy among the three. Therefore, we use only DTW in the remaining experiments, below.

**Experiment 2: Verification of LSPSO.** *An LSPSO algorithm was proposed in this paper. To verify its effectiveness, three classical bi-objective test functions (BNH, SRN and TNK) were selected. GD and SP were used as evaluation indicators. The number of populations, N was set to 100, and the number of iterations G = 250 was compared with MODE, NSGA-II and OMOPSO, respectively. Each algorithm ran independently 30 times for each test function, computing the mean and variance of GD and SP values. Analysis of Variance (ANOVA) [42] was used to test the significant difference of the GD and SP indicators between LSPSO and the other models (MODE, NSGA-II, and OMOPSO). In general, P values smaller than 0.05 indicate that there is a significant difference.*

Figures 3–5 show the solution obtained by LSPSO for BNH, SRN and TNK functions and the true Pareto frontier. The red circle is the optimal solution obtained by LSPSO, and the black line is the true Pareto front. The optimal solution sets of LSPSO are convergent and evenly distributed on the true Pareto front.

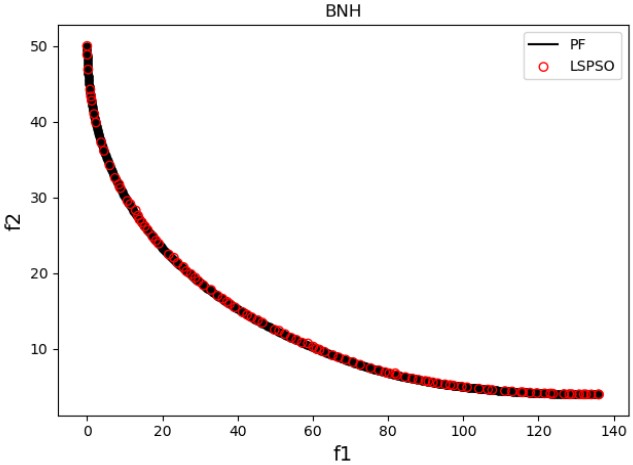

**Figure 3.** Pareto optimal solutions obtained by LSPSO for BNH.

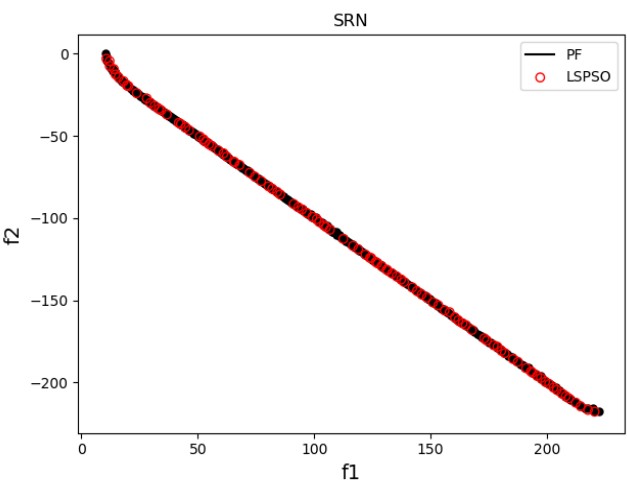

**Figure 4.** Pareto optimal solutions obtained by LSPSO for SRN.

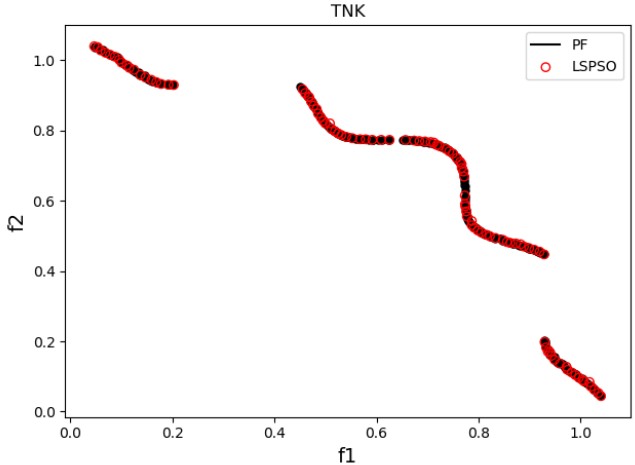

**Figure 5.** Pareto optimal solutions obtained by LSPSO for TNK.

Table 3 shows the GD and SP of MODE, NSGA-II, OMOPSO, and LSPSO in solving BNH, SRN, and TNK, respectively. Overall, our LSPSO is better than the other three methods according to the mean and std values of GD and SP.

**Table 3.** The GD and SP obtained by MODE, NSGA-II, OMOPSO and LSPSO in solving BNH, SRN and TNK.

| Function | | MODE | | NSGA-II | | OMOPSO | | LSPSO | |
|---|---|---|---|---|---|---|---|---|---|
| | | GD | SP | GD | SP | GD | SP | GD | SP |
| BNH | Mean | 0.5498 | 1.9369 | 0.1386 | 1.1174 | 0.1405 | 0.9745 | 0.1350 | 0.7127 |
| | Std | $8.77 \times 10^{-2}$ | 0.5502 | $1.56e \times 10^{-2}$ | $6.50 \times 10^{-2}$ | $1.46 \times 10^{-2}$ | $7.94 \times 10^{-2}$ | $1.36 \times 10^{-2}$ | $6.95 \times 10^{-2}$ |
| SRN | Mean | 2.8408 | 4.1774 | 0.6819 | 1.8991 | 0.5876 | 1.3269 | 0.4644 | 1.5033 |
| | Std | 0.4715 | 0.5601 | $9.02 \times 10^{-2}$ | 0.1593 | $6.58 \times 10^{-2}$ | 0.1020 | $3.77 \times 10^{-2}$ | 0.1149 |
| TNK | Mean | $4.96 \times 10^{-3}$ | $4.34 \times 10^{-2}$ | $2.45 \times 10^{-3}$ | $3.73 \times 10^{-2}$ | $2.68 \times 10^{-3}$ | $3.39 \times 10^{-2}$ | $2.41 \times 10^{-3}$ | $3.39 \times 10^{-2}$ |
| | Std | $6.91 \times 10^{-4}$ | $2.37 \times 10^{-3}$ | $2.24 \times 10^{-4}$ | $7.84 \times 10^{-4}$ | $2.44 \times 10^{-4}$ | $7.25 \times 10^{-4}$ | $1.46 \times 10^{-4}$ | $6.90 \times 10^{-4}$ |

When dealing with the BNH function, our LSPSO method outperforms the other three methods in the uniformity of the solution set, achieving a statistical significance of SP ($p < 0.001$). This can be observed from Figure 3, where the solution set of LSPSO is uniformly distributed on the Pareto front for BNH. The convergence of LSPSO is much better than that of MODE in term of GD ($p < 0.001$), but it is not significantly better than NSGA-II and OMOPSO ($p = 0.37$ and $p = 0.15$, respectively).

For the SRN function, the GD of LSPSO is significantly better than those on the other three methods ($p < 0.001$), indicating a good convergence. Regarding the uniformity of the solution set distribution (SP), although LSPSO is significantly better than MODE and NSGA-II ($p < 0.001$), it is worse than OMOPSO ($p < 0.001$). This maybe because the true Pareto frontier of the SRN function shows linearly distributed and global search has advantages when dealing with such problems, but when the true Pareto frontier of the test function is non-linear, our method can achieve much better results.

For the TNK function, the average and std of the GD and SP obtained by LSPSO are better than those on the other methods, indicating that LSPSO handles TNK well. According to the significance test, our LSPSO has similar performance with NSGA-II in terms of GD ($p = 0.48$), and with OMOPSO in terms of SP ($p = 0.95$).

In summary, LSPSO can better handle the bi-objective test functions, and provide effective support for the parameter optimization of DS. MODE, NSGA-II and OMPSO have strong global search capabilities and insufficient local search capabilities, so the solution sets obtained are not uniform and easily fall into a local optimum. The global and local search capabilities of PSO are mutually constrained and tend to fall into local optimum in the later stages of search. In this paper, a local search strategy is used to enhance the local search capability of the PSO, so that the improved PSO has independent global and local search capabilities. Therefore, the obtained solution set has better convergence and more uniform distribution.

**Experiment 3: Comparison of DSL performance with other SSTP methods.** *DSL is a combination of DTW + SVM (DS) and LSPSO. Define the space composed of the PL and the IS as the search space of the particle, and the prediction accuracy and efficiency set as the optimization objectives, LSPSO can compute the appropriate PL and IS that are set to the parameter of DS to predict SST. Here, we compare the performance of DSL with DTW, SVM, DS, LSTM in SSTP. In the situation of predicting 5-day SST with a given t-day of SST data, the DTW method is to find the most similar series of SST from historical data and take its following 5 days as the prediction. The SVM and LSTM both are trained by fitting nonlinear changes in SST. DS is to train a SVM by top-k similar series selected according to DTW.*

The comparison results are shown in Table 4. The average value of RMSE obtained by DS in predicting SST is 0.4468, which is lower than the average RMSE of the DTW. This indicates that the combination of DTW and SVM can effectively utilize the information after DTW mining. Secondly, the prediction results of the DS algorithm are better than those of SVM. This is because the SST sequence contains a lot of redundant information, which will interfere with the model during prediction, making the SVM prediction accuracy low.

**Table 4.** RMSE obtained when predicting SST by DTW, SVM, DS, LSTM and DSL.

| Station | DTW | SVM | DS | LSTM | DSL |
|---------|--------|--------|--------|--------|--------|
| 1 | 0.4866 | 0.7403 | 0.4640 | 0.5093 | 0.3255 |
| 2 | 0.5136 | 0.7117 | 0.4979 | 0.5075 | 0.4056 |
| 3 | 0.4606 | 0.7338 | 0.4387 | 0.4839 | 0.3334 |
| 4 | 0.4637 | 0.8383 | 0.4569 | 0.5458 | 0.3454 |
| 5 | 0.4544 | 0.6208 | 0.4259 | 0.4534 | 0.3862 |
| 6 | 0.4630 | 0.8694 | 0.4414 | 0.5710 | 0.4300 |
| 7 | 0.5204 | 0.8532 | 0.5121 | 0.5781 | 0.4816 |
| 8 | 0.4429 | 0.7840 | 0.3936 | 0.5154 | 0.2924 |
| 9 | 0.4161 | 0.7660 | 0.3911 | 0.5259 | 0.3534 |
| Avg | 0.4690 | 0.7686 | 0.4468 | 0.5211 | 0.3726 |

LSTM gains the average RMSE of 0.5211, which is worse than DTW, DS and DSL. Although LSTM is developed for dealing with long and short-term prediction problem, it does not work well in predicting SST. This is probably due to the non-stationarity of SST.

DSL obtains the optimal parameters of PL and IS by LSPSO, and uses them in DS to predict SST. Comparing the RMSE values between DS and DSL, it can be found that the performance of DSL prediction of SST is better than DS, indicating that the parameters of DS can be effectively optimized by LSPSO, which was responsible for a 16.7% improvement in prediction accuracy, in terms of reducing the RMSE. In summary, the overall effect of DSL prediction of SST is optimal, indicating that the method can effectively predict SST and verify the effectiveness of the proposed method.

We demonstrated the predicted results by different SSTP methods. Figure 6 shows a sample, randomly selected from the results. The black line represents the true values; the blue line represents the predicted results by using DTW; the red and yellow lines correspond to SVM and LSTM, respectively; the predicted results by using DSL are shown in green. It is clear that the results predicted by our method (green) are the closest ones to ground truth (black), including also the trend changes. On the other hand, the change trend of the other methods fluctuates significantly.

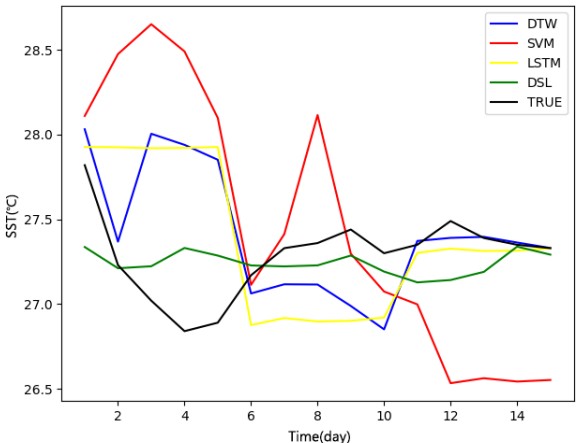

**Figure 6.** Comparison of SST prediction results by different methods, where black line is ground truth, green line is our model's prediction.

Finally, Figure 7 shows the operating efficiency of DS at predicting SST, before and after LSPSO optimization. RT represents the running time in seconds. The first nine columns provide RT for nine different SST sequences, with the average values given in the last column. The red histogram is the RT of the DS before optimization, and the blue histogram is the RT of DS after optimization. It can be clearly seen that, for each SST series, the RT before optimization is much longer, with an average 76% acceleration. Our results verify the effectiveness of the proposed method.

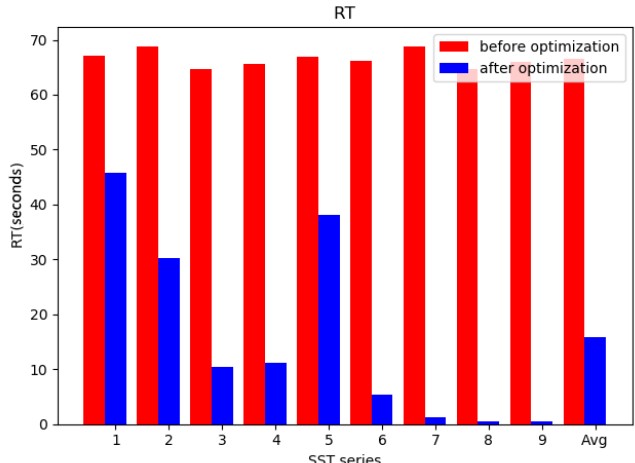

**Figure 7.** Comparison of RT before and after optimization.

## 5. Conclusions

This paper constructs a simple yet effective SSTP model, dubbed DSL, based on time-series similarity measure, multiple pattern learning and parameter optimization. We approach this complex issue through three stratagems: (1) Using Dynamic Time Warping (DTW) to mine the similarities in historical SST series. This can extract similar patterns of historical SST series simply and accurately. (2) Training an SVM using the top-k similar patterns, deriving a robust SSTP model that offers a 5-day prediction window based on multiple SST input sequences. Learning from multiple time-series sequences was instrumental to facilitating consistency enhancement and noise cancellation, thus achieving high prediction accuracy. (3) Developing an improved PSO method, dubbed LSPSO, to find the optimal parameters of the SSTP model, which uses a local search strategy to achieve the combined requirement of prediction accuracy and efficiency.

Our method strives for optimal model parameters (pattern length and interval step) and is suited for long-term series, leading also to significant improvements in SST trend predictions. The efficiency LSPSO was verified to have better convergence and more uniform distribution by comparing with different optimization methods such as MODE, NSGA-II and OMOPSO. With the LSPSO, the SST prediction using SVM and DTW achieved a 16.7% reduction in prediction error, at a 76% gain in operating efficiency. We also achieved a significant improvement in prediction accuracy of non-stationary SST time series compared to the more recent LSTM deep learning method. In general, our method provides a new way for marine operational forecasting.

In the work carried out so far, we have assumed that SST changes were affected only by internal factors. As a future extension, we shall strive to take into account also other external factors (such as air temperature, air pressure, etc.). This will require considering the correlations between SST and these factors. We plan to use association rules to analyze impact on SST, and then use multi-factor prediction models to predict SST, striving for more comprehensive and robust predictions.

**Author Contributions:** Conceptualization, Q.H. and W.S.; methodology, C.Z., W.S., C.P. and A.L.; software, C.Z.; investigation, Y.D.; resources, Z.H.; writing—original draft preparation, Q.H. and C.Z.; writing—review and editing, W.S., C.P. and A.L.; supervision, Q.H. and W.S. All authors have read and agreed to the published version of the manuscript.

**Funding:** The work is supported by the National Key R&D Program of China (2016YFC1401902), the National Natural Science Foundation of China (41671431), the Program for the Capacity Development for Shang Local Colleges (17050501900), the National Oceanic Administration Digital Ocean Science and Technology Key Laboratory Open Fund (B201801029) and the open fund of State Key Laboratory of Satellite Ocean Environment Dynamics, Second Institute of Oceanography, MNR (QNHX1913).

**Acknowledgments:** The sea surface temperature data were provided by State Key Laboratory of Satellite Ocean Environment Dynamics, the Second Institute of Oceanography, the Ministry of Natural Resources. The authors would also like to thank the anonymous reviewers for their very competent comments and helpful suggestions.

**Conflicts of Interest:** The authors declare that they have no known competing financial interests or personal relationships that could have appeared to influence the work reported in this paper.

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
