# Peer review of "Improved Particle Swarm Optimization for Sea Surface Temperature Prediction"

_energies, doi:10.3390/en13061369_

Round 1

Reviewer 1 Report

The authors of the presented manuscript delivers the goal set in it's title, "Improved Particle Swarm Optimization for Sea Surface Temperature Prediction". I consider that this manuscript contains valuable information and methods that can be applied in other fields.

However, some aspects needs to be resolved before publication. I invite the authors to follow the next observations:

  • please provide better quality graphs;
  • please add measurements units to the graphs axes;
  • it appears that a lot of hard ward was done to develop this manuscript, and yet it seems that the results are are not enough;
  • please provide better conclusions.

Best regards!

Author Response

  • We have provided higher quality graphs;
  • We have addedmeasurements units to the graphs axes
  • We have explained the relevant results in detail in the Results and Discussion section, highlighted in yellow
  • We have improvedconclusions, and marked them in yellow.

Thank you for your valuable suggestions.

Reviewer 2 Report

The paper is well written and technically sound.

The idea to use PSO with a Local Search Strategy has been adopted and applied by other researchers to cope with difficult optimization problems.

The authors should refer to some of these attempts in order to justify their decision to use a hybrid version of PSO with local search.

Among other contributions the authors should refer to the following:

  1. Tassopoulos, I.X., Beligiannis, G.N. Using particle swarm optimization to solve effectively the school timetabling problem. Soft Comput 16, 2012, pp. 1229–1252.
  2. A.A. Mousa, M.A. El-Shorbagy, W.F. Abd-El-Wahed, Local search based hybrid particle swarm optimization algorithm for multiobjective optimization, Swarm and Evolutionary Computation, Volume 3, 2012, pp. 1-14.
  3. Katsaragakis I.V., Tassopoulos, I.X., Beligiannis, G.N., A Comparative Study of Modern Heuristics on the School Timetabling Problem. Algorithms 2015, 8, 723-742.
  4. Lin, J.T., Chiu, C. A hybrid particle swarm optimization with local search for stochastic resource allocation problem. J Intell Manuf 29, 2018, pp. 481–495.

Author Response

In response to your question, we have referenced 2,4 as [31]and [32], the school timetabling problem is are not similar to ours.

Thank you for your valuable suggestions.

Reviewer 3 Report

The article entitled "Improved Particle Swarm Optimization for Sea Surface Temperature Prediction" is a well designed and correctly written work, which guides the reader through applied and developed algorithms step by step. Materials and methods section is clear and sound, all the methods used are widely described and appropriate literature is cited. 

The main drawback is the lack of discussion of obtained results with other works in the Results and discussion section. It would undoubtedly improve the quality of the presented article.  

Author Response

We have improved the section of results and discussion in two aspects: 1) discussion on the performance of MODE, NSGA-â…¡,OMOPSO and LSPSO; 2)discussion on the reasons for combining  local search strategy and PSO;3) discussion on the performance of DTW, SVM,LSTM and DSL;

Thank you for your valuable suggestions.